# A Validation Study for SHE Score for Acute Subdural Hematoma in the Elderly

**DOI:** 10.3390/brainsci12080981

**Published:** 2022-07-26

**Authors:** Martin Vychopen, Motaz Hamed, Majd Bahna, Attila Racz, Inja Ilic, Abdallah Salemdawod, Matthias Schneider, Felix Lehmann, Lars Eichhorn, Christian Bode, Andreas H. Jacobs, Charlotte Behning, Patrick Schuss, Erdem Güresir, Hartmut Vatter, Valeri Borger

**Affiliations:** 1Department of Neurosurgery, University Hospital Bonn, 53127 Bonn, Germany; motaz.hamed@ukbonn.de (M.H.); majd.bahna@ukbonn.de (M.B.); inja.ilic@ukbonn.de (I.I.); abdallah.salemdawod@ukbonn.de (A.S.); matthias.schneider@ukbonn.de (M.S.); patrick.schuss@ukb.de (P.S.); erdem.gueresir@ukbonn.de (E.G.); hartmut.vatter@ukbonn.de (H.V.); valeri.borger@ukbonn.de (V.B.); 2Department of Epileptology, University Hospital Bonn, 53127 Bonn, Germany; attila.racz@ukbonn.de; 3Department of Anesthesiology and Intensive Care Medicine, University Hospital Bonn, 53127 Bonn, Germany; felix.lehmann@ukbonn.de (F.L.); lars.eichhorn@ukbonn.de (L.E.); christian.bode@ukbonn.de (C.B.); 4Department of Geriatric Medicine and Neurology, Johanniter Hospital Bonn, 53113 Bonn, Germany; andreas.jacobs@bn.johanniter-kliniken.de; 5Department of Medical Biometry, Informatics and Epidemiology, University Hospital Bonn, 53127 Bonn, Germany; behning@imbie.uni-bonn.de

**Keywords:** subdural, hematoma, acute, nonconvulsive, status, epilepticus

## Abstract

Objective: The aim of this study was the verification of the Subdural Hematoma in the Elderly (SHE) score proposed by Alford et al. as a mortality predictor in patients older than 65 years with nontraumatic/minor trauma acute subdural hematoma (aSDH). Additionally, we evaluated further predictors associated with poor outcome. Methods: Patients were scored according to age (1 point is given if patients were older than 80 years), GCS by admission (1 point for GCS 5–12, 2 points for GCS 3–4), and SDH volume (1 point for volume 50 mL). The sum of points determines the SHE score. Multivariate logistic regression analysis was performed to identify additional independent risk factors associated with 30-day mortality. Results: We evaluated 131 patients with aSDH who were treated at our institution between 2008 and 2020. We observed the same 30-day mortality rates published by Alford et al.: SHE 0: 4.3% vs. 3.2%, *p* = 1.0; SHE 1: 12.2% vs. 13.1%, *p* = 1.0; SHE 2: 36.6% vs. 32.7%, *p* = 0.8; SHE 3: 97.1% vs. 95.7%, *p* = 1.0 and SHE 4: 100% vs. 100%, *p* = 1.0. Additionally, 18 patients who developed status epilepticus (SE) had a mortality of 100 percent regardless of the SHE score. The distribution of SE among the groups was: 1 for SHE 1, 6 for SHE 2, 9 for SHE 3, and 2 for SHE 4. The logistic regression showed the surgical evacuation to be the only significant risk factor for developing the seizure. All patients who developed SE underwent surgery (*p* = 0.0065). Furthermore, SHE 3 and 4 showed no difference regarding the outcome between surgical and conservative treatment. Conclusions: SHE score is a reliable mortality predictor for minor trauma acute subdural hematoma in elderly patients. In addition, we identified status epilepticus as a strong life-expectancy-limiting factor in patients undergoing surgical evacuation.

## 1. Introduction

Acute subdural hematoma (aSDH) is one of the most common pathologies in neurosurgical care. The general prevalence is increasing with the growth of the aging population. SDH is expected to become one of the most common neurosurgical conditions by 2030 [1]. In particular, aSDH often presents a condition with poor clinical outcome and recovery regardless of the treatment [2]. Currently, there are no guidelines available to guide neurosurgeons during the decision making process to identify patients with the best chance for recovery after surgical treatment. Therefore, the identification of patients who profit from surgical intervention is crucial. However, there is a known correlation between poor outcome and age with reported highly mortality rates in elderly patients [3,4,5]. In a prospective observational study reported by Weimer et al. [6] on a mixed population of 116 patients with SDH, aside from old age, poor premorbid neurological status, admission Glasgow Coma Score (GCS), and history of smoking and fever during hospitalization were independent predictors for poor outcome. There are several series providing scores to predict the outcome in patients with SDH [7,8,9,10,11,12,13]. Apart from the abovementioned well-known outcome predictors, status epilepticus is an unrecognized prognostic factor in patients with surgical interventions for SDH [14,15]. The majority of these reports deal with heterogeneous patient populations consisting of patients with aSDH, chronic SDH, and mixed SDH.

Recently, Alford et al. [4] proposed a highly specific subdural hematoma in the elderly (SHE) score using admission characteristics to predict the 30-day mortality. Those characteristics include age, Glasgow coma scale (GCS) by admission, and aSDH volume. In original publication, Alford et al. [4] tried to develop a scoring system for all types of subdural hematoma; chronic, mixed, and acute. However, the proposed SHE score showed the highest discriminative ability in patients with acute subdural hematoma. Despite lower specificity and sensitivity of the SHE score in chronic subdural hematoma in original study, Petrella et al. already performed a SHE-Score evaluation for patient with chronic subdural hematoma [16].

In this study, we performed a single center verification of the SHE score exclusively for patients with aSDH in different clinical settings. Additionally, we evaluated the impact of status epilepticus on the 30-day mortality among those patients.

## 2. Materials and Methods

We retrospectively evaluated a cohort of 209 consecutive patients with aSDH treated at our institution during a period from December 2007 to June 2020. Excluded were patients who were younger than 65 years (*n* = 48), patients with severe traumatic brain injury (polytrauma and high speed impact injury and open head injury) (*n* = 22), patients with aneurysmatic subarachnoidal hemorrhage (*n* = 6) and patients with dural AV fistula as underlying pathology (*n* = 2) leaving 131 patients with minor-trauma aSDH suitable for the analysis. We defined minor-trauma head injury as a direct head impact without prolonged loss of consciousness and absence of open injury/skull fracture and initially without clinical deterioration/or focal neurological deficit [17,18].

Additionally, we investigated age, GCS on admission, the use of anticoagulants, the length of hospital stay, and 30-day clinical outcome according to Glasgow outcome scale (GOS).

### 2.1. SHE Score, Treatment Algorithm, and Primary Endpoint

All patients underwent a CT scan at admission, from which the aSDH volume was calculated as previously described by Sucu et al. [19]. All patients with a hematoma volume equal or greater than 50 mL were given 1 point. The GCS at admission to the neurosurgical ward was scored as 0 for GCS 13–15, 1 for GCS 5–12, and 2 for GCS 3–4. All patients aged over 80 years obtained 1 point. The sum of all points determined the SHE score as reported by Alford et al. [4]. For detailed information, see Table 1.

For surgery indication, treatment algorithm proposed by Bullock et al. was followed [7]. In patients with neurological deficit, consciousness alteration, or in those where the thickness of the hematoma overreached the thickness of the skull and caused clinically relevant mass effect accompanied by midline shift, craniotomy and hematoma evacuation were performed. All craniotomies were performed in a standardized fashion. A single burr hole is placed 4 cm superiorly from the meatus acusticus externus and 6 × 6 cm bone flap is created. After hematoma evacuation, up to two subdural drains might be placed to drain the remaining blood. The decision to place the drain was based on the judgement of attending surgeon. Because of the retrospective data collection, we were not able to evaluate the amount of drained fluid.

We strived to achieve postoperative extubation as early as possible.

Additionally, 30-day mortality, clinical outcome according to Glasgow outcome score (GOS), and complications occurred during the hospital stay including the development of focal seizure, generalized seizure [20], and nonconvulsive status epilepticus (ncSE) [21] were noted.

### 2.2. Nonconvulsive Status Epilepticus—Secondary Endpoint

Patients, whose extubation failed because of the consciousness state, received an electroencephalography (EEG) to identify those with nonconvulsive status epilepticus (ncSE). The postoperative CT-scan was performed 24 h after hematoma evacuation and subsequently, the subdural drains were removed.

In patients who did not show adequate neurological response after reduction of sedation drugs, a continuous electroencephalography (cEEG) was performed to rule out the nonconvulsive status epilepticus (ncSE). Patients with ncSE received an aggressive treatment as described by Fergusson et al. [22]

If the EEG finding was positive, levetiracetam was used as a therapy of choice and patient received (cEEG). If the ncSE persisted despite the monotherapy, lacosamid and briviracetam were added successively in an attempt to control the seizure. Finally, patients with super-refractory ncSE received isoflurane under cEEG to induce burst supression [23]. All EEG recordings were analyzed by board of certificated and experienced neurologists. The treatment of ncSE was adopted on individual conditions and EEG findings in each patient.

Furthermore, patients were divided according to low (SHE 0–2) and high (SHE 3–4) SHE score. Subsequently, we analyzed the correlation of the score with ncSE.

### 2.3. Surgery vs. Conservative Treatment

Finally, we compared the patients according to the therapy of choice for the SDH and divided them into two groups: the best conservative therapy vs. surgical evacuation followed by conservative care.

### 2.4. Statistics

Fisher’s exact test and chi-square test were performed to compare the results. Yates correction was used for all chi-square tests to avoid type 1 error. All values with *p* 0.05 were considered as significant. Subsequently, logistic regression analysis was used for identification of risk factors for development of ncSE presented by Won et al. Additionally, we performed a receiver operating characteristics (ROC) evaluation to examine the sensitivity and specificity of SHE for predicting 30-day mortality [16,24]. We then used the ROC curve comparison method published by Hanley et al. to compare the difference between the areas under two independent ROC curves to verify the SHE score [15].

## 3. Results

### 3.1. Validation of SHE Score in the Elderly with Acute Subdural Hematoma

At the author’s institution, 209 patients with spontaneous or minor trauma aSDH were hospitalized between December 2007 and June 2020. Excluded were patients who were younger than 65 years (*p* = 48), patients with severe traumatic brain injury (polytrauma and high speed impact injury and open head injury) (*p* = 22), patients with aneurysmatic subarachnoidal hemorrhage (*p* = 6), and patients with dural AV fistula as underlying pathology (*p* = 2). A total of 131 patients were left for the analysis. Mean age was 78.4 ± 7.4 years with male to female ratio of 1.15. For patient characteristics, see Table 2.

The retrospective SHE scoring system was used to evaluate the patients by outcome. According to the SHE score, the 30-day mortality rate for SHE score of 0 was 4.3% and increased for SHE score of 1, 2, 3, and 4 to 12.1%, 26.6%, 97.0%, and 100%, respectively (Table 3). We compared our results with the results published by Alford et al. [4]. We did not find any statistically significant differences between the two patient populations. The functional outcome at day 30 is shown in Figure 1.

A favorable functional outcome (GOS 4–5) was achieved in (96%) of the patients with SHE score 0 and in (76%) in patients with SHE score 1, respectively. Patients with SHE score 2 show only in 57% a favorable functional outcome. None of the patients with SHE score 3 and 4 achieved a favorable outcome. For detailed information, see Figure 1.

We performed a ROC evaluation to examine the sensitivity and specificity of SHE for predicting 30-day mortality. Our results confirmed high sensitivity (98.59% with CI (95%) of 92.4–99.6) and high specificity (73.3% with CI (95%) of 60.3–83.9) of the score without any significant difference compared to data published by Alford et al. [4] (sensitivity 94.2% with CI (95%) of 90.2–96.9 and specificity of 51.52% with CI (95%) of 38.9–64.0).

The ROC evaluation showed the area under curve (AUC) of 0.899 with CI (95%) of 0.842–0.957 compared to 0.941 of Alford et al. [4]. The comparison showed no statistically significant difference between the curves (*p* = 0.13). For detailed information, see Figure 2.

### 3.2. Nonconvulsive Status Epilepticus as Strongest Independent Additional Outcome Measure

In our study cohort, we found 18 patients (14%) who developed nonconvulsive status epilepticus (ncSE) with the necessity of multidrug antiepileptic therapy. All these patients died within 30 days of therapy. In comparison to patients without ncSE, the overall 30-day mortality rate was significantly higher in patients with diagnosed ncSE (100% vs. 37.1%; *p* 0.0001). In logistic regression analysis, the surgical evacuation of the hematoma was the only identified independent risk factor for development of the seizure (*p* = 0.015, CI 95% 1.44–29.55, OR = 6.52). In univariate analysis, we found no association between the drain insertion and development of ncSE (*p* = 0.2039). For detailed information, see Table 4.

Regarding focal seizures, we found no outcome limitation in these patients. Overall, 10 patients were identified, who underwent the antiepileptic therapy due to isolated seizures. All of them survived the 30-day period. Distribution of ncSE among the SHE groups All patients with ncSE underwent surgical evacuation of the hematoma (*p* = 0.0065). Patients with a high SHE score were more likely to develop ncSE compared to those with a low SHE score (*p* = 0.015). The limitation of this statement is the small number of patients with ncSE in each group. For detailed information, see Table 3.

### 3.3. Surgery vs. Conservative Treatment

Among the study population, surgical evacuation of the hematoma was performed in 99 out of 131 patients (75.5%). There was no significant difference regarding the functional outcome between patients undergoing surgical evacuation of the hematoma vs. patients with conservative treatment having SHE score of 0, 3, and 4. The sample size in the group of the patients with best conservative treatment and SHE score of 1 and 2 was too small to perform reliable statistical analysis. For detailed information, see Table 2.

We found no statistical correlation between surgery indication and SHE score (*p* = 0.3998).

## 4. Discussion

### 4.1. SHE Score—Primary Endpoint

In this series, an external validation of SHE score proposed by Alford et al. [4] was performed to evaluate its accuracy to predict 30-day mortality in the elderly suffering from aSDH and the perspective use of this scoring system in routine clinical setting. We presentexternal validation of the SHE score for minor trauma aSDH in our clinical setting. Our dataset confirms the high specificity of the score for 30-day mortality.

In general, predictors of poor clinical outcome in elderly patients suffering from aSDH were extensively described [7,8,9,10,11,12,13]. Patients suffering from traumatic aSDH are known to have higher risk of developing SE associated with poor outcome [25,26].

Regarding surgery as a risk factor for SE, the appropriate indication to hematoma evacuation seems to play a crucial role in aSDH therapy. In a large retrospective population study, only 7.5% of aSDH-patients with the best conservative treatment had to undergo secondary indicated surgical evacuation [27]. Our data suggests that patients scored with SHE 0 profit from the best conservative treatment leading to secondary chronification of the hematoma [7,28]. The only noted death in our SHE 0 group was due to non-neurosurgical comorbidities resulting in cardiac failure.

Because of the pronounced symptoms, patients with SHE score 1 and SHE score 2 were almost always indicated for surgery. Unfortunately, a small control group of patients with conservative treatment in our patient cohort does not allow statistical evaluation. A lower mortality rate for SHE score 1 (16%) and for SHE score 2 (34.5%) suggests that patients in those groups benefit from surgery. However, despite the good outcome the development of non-convulsive SE is a prognosis-limiting factor that should lead to careful preoperative evaluation and accurate operative indication.

Patients in SHE score 3 and SHE score 4 group showed generally poor outcome regardless of the therapy of choice. Our data, as well as the data published by Alford et al. [4] contains in total only 3 patients in SHE score 3 group, who survived the 30-day period, 2 of them with poor GOS. The profit of surgical evacuation of the hematoma in these particular patients seems to be questionable. According to our data as well as to data published by Alford et al. [4], the surgery does not seem to improve clinical outcome.

In our dataset, we did not use SHE score prospectively for surgical indication. The data assessment was strictly retrospective, which is why a bias in surgical indication was expected. Nevertheless, if divided in low SHE score (SHE 0–2) and high SHE score (SHE 3–4), there was no bias in surgery indication (*p* = 0.3998).

To sum up, the SHE score proved to be a reliable and easy-to-use 30-day outcome predictor similar to already published and clinically widely utilized ICH score [29].

### 4.2. ncSE—Secondary Endpoint

Additionally, we identified development of ncSE postoperatively as an independent risk factor associated with 100% mortality regardless of the initial SHE score. Furthermore, the higher SHE score correlates positively with the development of ncSE.

Our data also suggest that patients who undergo surgical treatment are at higher risk of developing ncSE.

Surprisingly, we did not find any association between development of ncSE and insertion of subdural drain (*p* = 0.2039). Therefore, being statistically insignificant in univariate analysis, the drain insertion was not included in our multivariate analysis.

As previously described, the development of ncSE dramatically increases the risk of a poor cognitive and neurological outcome in adults [30], as well as in neurological critical care patients with ICH [31]. Because the SDH is supposed to be the underlying condition for the ncSE, an aggressive antiepileptic drug treatment is performed in accordance with already published data [22,32].

Although the number of patients developing ncSE is higher in SHE 3 and 4, the implementation of ncSE as an item in the SHE score is not possible because of the preclinical character of the SHE score. The ncSE is a condition developed solely in the course of the treatment and was an outcome-limiting condition regardless of the SHE score value.

In comparison to ncSE, the development of isolated seizures does not seem to affect the mortality. All patients with isolated seizures in our study cohort received antiepileptic monotherapy (levetiracetam or lacosamide) to achieve seizure-free status. In these patients, the outcome was affected only by their SHE score. In accordance to already published data, isolated seizures did not affect the neurological outcome in critically ill patients [33,34].

Compared to Ali Seifi et al. [35] (0.5%), our cohort showed higher incidence of status epilepticus. This might have two possible explanations. In accordance with German neurological society [36], our institutional therapy algorithm does not include use of antiepileptic prophylaxis, as proposed by Won et al. [37] This might be the reason for relatively high incidence of ncSE in our cohort. However, our aggressive diagnostic protocol with low-threshold EEG, i.e., continuous EEG might also lead to higher number of diagnostically verified ncSE. In conclusion, we cannot make a statement about either the benefits of prophylactic use of antiepileptic drugs or the incidence of ncSE. A prospective evaluation should be performed to specifically address these questions.

## 5. Limitations

This study has several limitations. As the original publication did not provide specific epidemiological data, we were not able to statistically compare both populations. Due to the retrospective design, it is highly probable not to recruit patients treated conservatively, mainly for SHE scores 1 and 2. The nonrandomized setting left the therapy of choice in the decision of the attending physician. The SDH volume determination method (ABC/2)5 might cause a systematic error in results (up to 5% error according to Konczalla et al. [38]. Although all patients received standardized treatment, retrospective design and the absence of study protocol might further corrupt the dataset. Due to the retrospective design of the study, we were not able to evaluate long-term outcome. We also expect a sample bias in the SHE 0 and 1, where asymptomatic patient are frequently treated in an outpatient setting without being directly admitted to neurosurgery. Due to single-center study design and rarity of the diagnosis, only 18 patients with nonconvulsive status epilepticus were identified for the analysis. Due to use of low-threshold EEG diagnostics and omittance of prophylactic antiepileptic drugs in our institutional algorithm, we cannot evaluate the possible pros and cons of antiepileptic prophylaxis by patients with mild TBI.

## 6. Conclusions

### 6.1. SHE Score—Reliable Mortality Predictor

The external validation of the SHE score was confirmed for aSDH. The SHE score is a strong mortality predictor for patients with minor trauma aSDH and has a high clinical utility and applicability to support the decision making process by the clinicians dealing with this difficult to treat patients. We think that patients with lower SHE score might profit from conservative treatment if clinically tenable. Furthermore, patients with high SHE score (3–4) and major medical comorbidities might be considered for best supportive care due to poor outcomes in this subgroup. This statement is strongly limited by retrospective setting of our study. The decision for “restrictive surgical approach” should be made solely on individual basis.

### 6.2. ncSE—Secondary Endpoint

Our data identified nonconvulsive status epilepticus as a strong life-expectancy-limiting factor in patients undergoing surgical evacuation. However, attention should be paid to patients with higher SHE score, who were surgically treated with respect to identification of ncSE, especially in cases with clinical deterioration following surgery. Identification of EEG-based measures early postoperatively predicting development of ncSE would be highly desirable to enable early aggressive antiepileptic drug therapy. The possible implication of our clinical finding might be a careful consideration of therapy goal by patients with high SHE score (3–4) who postoperatively develop ncSE. Our data suggest 100% mortality in this subgroup of patients. In such case, a multidisciplinary evaluation of therapy goal should be performed. The major limitation of this statement is the lack of statistical power (in total, only 18 patients with ncSE were identified).

## Figures and Tables

**Figure 1 brainsci-12-00981-f001:**
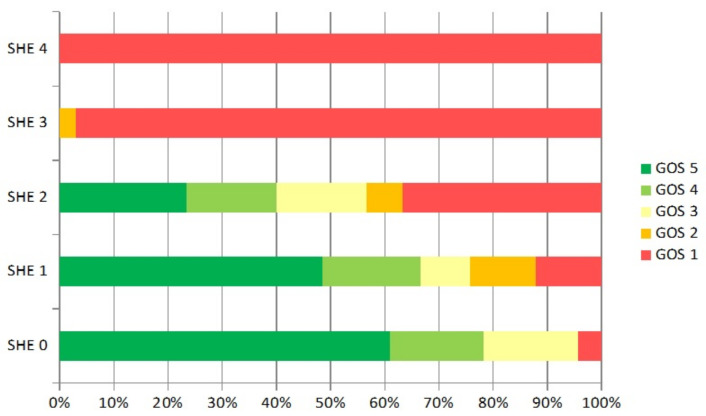
(**a**) Outcome according to Glasgow outcome score (GOS) at day 30. Positive outcome for SHE 0 = 96% and decreases for SHE 1, 2, 3, and 4 with 76%, 57%, 0%, and 0%, respectively.

**Figure 2 brainsci-12-00981-f002:**
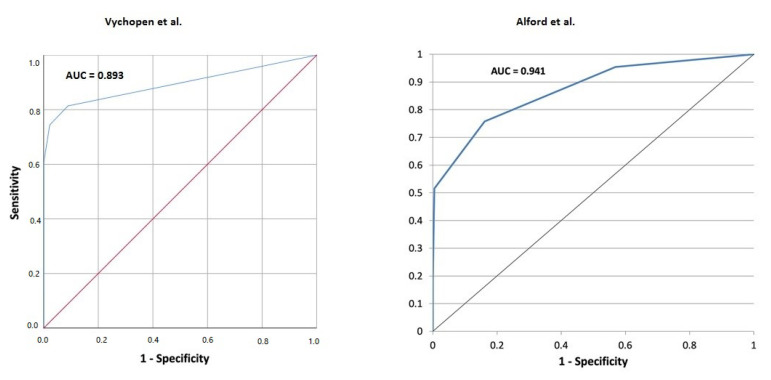
ROC curve comparison. There is no statistically significant difference between both curves (Vychopen AUC = 0.893, CI 0.839–0.948; Alford AUC = 0.941; *p* = 0.1856) [4]. Both curves showed good predictive capacity of SHE score.

**Table 1 brainsci-12-00981-t001:** SHE Score system moddified after Alford et al. [4]: GCS: Glasgow coma scale, mL: milliliter.

Variable	Points Obtained
Age 80 years	1
GCS 15–12	0
GCS 12–5	1
GCS 4–3	2
Hematoma volume 100 mL	1

**Table 2 brainsci-12-00981-t002:** Patient characteristics: n: number of patients, SD: standard deviation, mL: milliliter.

Patient Characteristics			
Number of patients		131	
Sex (male:female)		68:63	
Mean age (years ± SD)		78.4 ± 7.4	
Mean hematoma size (mL)		45	
History of anticoagulants/antiplatelet medication, n (%)		103 (79)	
Focal seizure, n (%)		10 (7)	
Generalized seizure, n (%)		18 (14)	
30-day mortality, n (%)		60 (46)	
Surgical evacuation, n (%)		99 (76)	
**Surgical evacuation vs. conservative treatment**			
**SHE Score**	**Surgical evacuation**	**Conservative treatment**	* **p** * **values**
Mean age (years ± SD)	78.2 ± 7.2	79 ± 7.4	0.5882
Sex (male:female)	55:44	13:19	0.1588
GCS by admission ± SD	10.2 ± 4.6	9.5 ± 5.2	0.4700
Focal seizure (%)	8 (8)	2 (6)	0.0065
Generalized seizure (%)	18 (18)	0 (0)	0.7345
30-day mortality (%)	45 (45)	15 (47)	0.99
SHE 0	14	9	
30-days mortality	0 (0%)	1 (11.1%)	0.3577
SHE 1	24	9	
30-days mortality	4 (16.6%)	0 (0%)	
SHE 2	29	1	
30-days mortality	10 (34.5%)	1 (100%)	
SHE 3	26	8	
30-days mortality	25 (96.1%)	8 (100%)	0.99
SHE 4	6	5	
30-days mortality	6 (100%)	5 (100%)	0.99

**Table 3 brainsci-12-00981-t003:** Comparison Vychopen vs. Alfort, 30-day mortality and distribution of nonconvulsive status epilepticus in correlation with SHE score. SHE: subdural hematoma in elderly score.

Comparison Vychopen vs. Alford 30-Day Mortality			
SHE Score	Vychopen 30-Day Mortality	Alford 30-Day Mortality	*p* Value
SHE 0	1/23 (4.3%)	3/94 (3.2%)	0.7135
SHE 1	4/33 (12.1%)	13/99 (13.1%)	0.8807
SHE 2	11/30 (36.6%)	16/49 (32.7%)	0.9039
SHE 3	33/34 (97%)	22/23 (95.6%)	0.7771
SHE 4	11/11 (100%)	12/12 (100%)	0.99
**Distribution of nonconvulsive Status epilepticus in correlation with SHE score.**			
**SHE score**	**Nonconvulsive** **status epilepticus**	**%**	
SHE 0	0/23	0	
SHE 1	1/33	3	
SHE 2	6/30	20	
SHE 3	9/34	26.5	
SHE 4	2/11	18.2	

**Table 4 brainsci-12-00981-t004:** Logistic regression analysis of risk factors for the seizure. OR—odds ratio, CI 95%—confidence interval 95%, SHE subdural hematoma in elderly score.

Variable	OR	CI 95%	*p* Value
Anticoagulation	0.71	0.24–2.12	0.55
SHE Score	1.055	0.74–1.50	0.76
Hematoma evacuation	6.52	1.44–29.55	0.015

## Data Availability

The datasets generated during and/or analyzed during the current study are available from the corresponding author on reasonable request.

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
