# Peer review of "A Validation Study for SHE Score for Acute Subdural Hematoma in the Elderly"

_brainsci, 2022, doi:10.3390/brainsci12080981_

Round 1

Reviewer 1 Report

Thank you for reaching out to review this work.

The authors aimed to perform an external validation study of the SHE score (Subdural hematoma in the elderly score).

For this purpose, they used a monocentric retrospective patient series.

General remarks

First, I want to congratulate the authors for conducting this external validation study for the SHE score originally described by Alford and colleagues in 2019 (Alford EN, Rotman LE, Erwood MS, Oster RA, Davis MC, Pittman HBC, Zeiger HE, Fisher WS. Development of the Subdural Hematoma in the Elderly (SHE) score to predict mortality. J Neurosurg. 2019 Apr).

Surgical decision is never easy to make in face with an old patient suffering from a subdural hematoma. This is particularly true in the case of acute subdural hematoma (ASH) which is a much more aggressive disease for old patients compared to mixed or chronic subdural hematoma, with a high degree of pre-operative uncertainty regarding the long-term functional outcome (Baucher G et al., Predictive Factors of Poor Prognosis After Surgical Management of Traumatic Acute Subdural Hematomas: A Single-Center Series. World Neurosurg. 2019 Jun).

A few years ago, Alford and colleagues provided an easy-to-use clinical tool aiming to assist the neurosurgeon in the surgical indication in face with old patients suffering from subdural hematoma. This work was warmly received by the scientific community (14 citations in the past three years) but remained quite unknown from the general neurosurgical community. An external validation study demonstrating the usefulness of this score would allow a wider day-to-day clinical use of this precious tool. Besides, Vychopen and colleagues focused their study solely on the elderly suffering from ASH, whereas the SHE score was originally described for all subtypes of subdural hematoma.

After careful consideration, it seems only logical to re-assess the validity of the SHE score for ASH only. Indeed, ASH may be directly fatal for old patients, whereas patients operated on for chronic or mixed subdural hematoma usually overcome this neurosurgical challenge; and only a fraction of them may die from bedsore complications afterwards.

Major comments

1. Title

The title used by the authors seems too lengthy and makes it difficult for the reader to immediately embrace the purpose of this work:

“SHE Score as reliable predictor of outcome with status epilepticus as independent mortality factor in elderly patients with acute subdural hematoma”

A simple title such as “A validation study for the SHE score in the elderly with acute subdural hematoma” would also do the job, don’t you think? This is just a proposition and the final decision must come from the authors.

Last, I think that the analysis concerning the status epilepticus-related mortality is confusing here and should not appear in the title.

2. Methods

2.1 Comparability between the original study by Alford (2019, JNS) and the current manuscript of Vychopen

The inclusion criteria used by Alford were as follows:

“The target population is elderly patients (> 65 years of age) who have a diagnosis of SDH and have a history of no or only minor trauma. Patients may have acute, chronic, or mixed-density  SDH“

The exclusion criteria used by Alford were as follows:

“The SHE score is not intended to be used for patients who have sustained a high-velocity trauma, defined as any motor vehicle collision, pedestrian struck, or fall from a height of 10 feet or greater”

The exclusion criteria used by Vychopen as stated in Materials and Methods were as follows:

“patients with subarachnoidal 51 hemorrhage (n=6) and patients with dural AV-fistula as underlying pathology”

The patients excluded in the Results section were then described as follows:

“accompanying intracerebral hemorrhage (n=9)

 Thus, I have two remarks:

-Could you be more specific regarding this exclusion criteria? Do you want to exclude patients with associated traumatic subarachnoid hemorrhage, or patient with any kind of associated traumatic subarachnoid hemorrhage or intraparenchymal traumatic brain injury?

-I wonder why these patients were excluded is they only sustained none or minor head trauma?

2.2 Treatment algorithm

The authors use precise clinical and radiologic criteria for the surgical indication of acute subdural hematoma. These criteria have been summarized in recommendations that could be reminded here:

- Bullock MR, Chesnut R, Ghajar J, Gordon D, Hartl R, Newell DW, Servadei F, Walters BC, Wilberger JE; Surgical Management of Traumatic Brain Injury Author Group. Surgical management of acute subdural hematomas. Neurosurgery. 2006 Mar;58(3 Suppl):S16-24; discussion Si-iv. PMID: 16710968.

2.3 Primary and secondary endpoints

Validation of the SHE score and ncSE-related mortality are both assessed as a primary endpoints.

This being said, the main point of this article is to perform an external validation of the SHE score, which is a pre-operative prognostic score. ncSE appears as a critical post-operative criterion of worse prognostic.

This, I think that the assessment of the impact of ncSE on the 30-days mortality should constitute a secondary endpoint.

Results

3.1 Comparability between populations

A general epidemiological table displaying:

-Epidemiological data concerning the population included in Vychopen study,

-With comparison between the patients operated on and those who did not undergo surgery,

-And compared to the epidemiological data of the original work of Alford regarding patients with acute subdural hematoma (Chi-square…),

Would be welcome in this external validation study.

3.2 SHE score

A table reminding the SHE score would be welcome.

3.3 Tables

This manuscript contains 7 tables.

The authors should try to reduce this number and to reorganize relevant data into two or three main tables, for example one epidemiological table (see remark 3.1), one results table, and one table for the comparability of the SHE score between the original work by Aflord and this manuscript. This is just a proposition and the authors should reorganize the results data according to their own judgement.

Last, I don’t see the usefulness of Table 4. The text already summarizes such information sufficiently.

3.4 Non-convulsivant status epilepticus (ncSE)

The authors find a strong correlation between surgical evacuation of ASH, the occurrence of post-operative ncSE, and 30-days mortality (100%). What is more, none of the patients who were not operated on developed ncSE.

Nevertheless, I am not sure that one can state that surgery is responsible for the occurrence of ncSE based on these data. I think that the following points can, at least in part, help to understand such a result:

-Probable lack of power, as stated by the authors. Only 18 ncSE patients left for analysis. This should be highlighted.

-The surgical indication. Were more patients operated on in the high SHE group compared with the low SHE group?

-Surgical protocol. In our institution, the insertion of a subdural drain is systematic after the surgical evacuation of chronic subdural hematoma but exceptional after ASH. Besides, previous studies have found that drain insertion may be a risk factor for developing status epilepticus in chronic subdural hematoma (Won SY et al., Seizure and status epilepticus in chronic subdural hematoma. Acta Neurol Scand. 2019 Sep).

-The seriousness of the neurological status of patients who developed ncSE (based on the SHE score, Table 6), although this does not appear significant in Table 5.

-The aggressiveness of the resuscitation protocol against ncSE.

However, I do agree with the authors that ncSE is a very serious condition, which may even prove fatal in many cases even with aggressive medical management. It appears essential for this major finding to be highlighted. In my opinion, it should constitute the secondary endpoint of the study.

3.5 Institutional habits regarding epilepsy management

Do the patients treated for ASH at your institution receive anti-epileptic drugs in a systematic manner, whether or not they are operated on? This could, at least in part, explain the high rate of ncSE (18/131, 13.7%) in this cohort.

Indeed, there are level 1 / level 2 recommendations that support the systematic use of anti-epileptic drugs in patients suffering from ASH with cumulative risk factors:

- Won SY, Konczalla J, Dubinski D, Cattani A, Cuca C, Seifert V, Rosenow F, Strzelczyk A, Freiman TM. A systematic review of epileptic seizures in adults with subdural haematomas. Seizure. 2017 Feb;45:28-35. doi: 10.1016/j.seizure.2016.11.017. Epub 2016 Nov 25. PMID: 27914224.

3.6 Surgical protocol

Could you provide the amount of fluid collected in the subdural drains during the 24-hours drainage period? If there is only a small volume of fluid collected, this could open the discussion about the usefulness of systematic subdural drainage.

Indeed, although subdural drainage is necessary in chronic subdural hematoma surgery, I am not sure whether it has proven such utility in the case of ASH (Santarius T et al., Use of drains versus no drains after burr-hole evacuation of chronic subdural haematoma: a randomised controlled trial. Lancet. 2009 Sep).

This remark is based on the fact that subdural drain may be a risk factor for developing ncSE (Won SY et al., Seizure and status epilepticus in chronic subdural hematoma. Acta Neurol Scand. 2019 Sep).

4. Discussion

4.1 Surgical indication.

I think that the authors should be very careful when they say that because surgery may be a risk factor for developing ncSE, surgical benefits should be carefully weighted against the potential risks. There is probably a major bias related to a lack of power when you perform a subgroup analysis with only 18 patients (beta risk).

Although I don’t have the precise analysis on hand, a lot of old patients operated on for ASH in our institution simply don’t make it because of the major level of pre-operative brain compression on pre-operative CT scan, without any evidence of ncSE on the EEG. This is mainly related to the pre-operative GCS score (included in the SHE score) and the presence of a fixed dilated pupil (not included in the SHE score) in such a fragile population.

There already are clear surgical indications for the evacuation of ASH (Bullock MR et al., Surgical Management of Traumatic Brain Injury Author Group. Surgical management of acute subdural hematomas. Neurosurgery. 2006 Mar). I agree with the fact that these recommendations tell us “Technically, this ASH should be operated on” but not “after the surgery, you patient will do better”. This is precisely the aim of the SHE score. From my point of view, the purpose of the present manuscript is to validate this score with a second external cohort of patients, and not to warn surgeons regarding the potential risk of post-operative epilepsy based on a subgroup analysis of 18 patients.

4.2 Secondary endpoint

Hence, the data regarding ncSE in this patient series should be considered as a secondary endpoint. ncSE is already informally considered by clinicians as a predictive factor of poor functional outcome, but to my knowledge, it has never been studied as such in the case of surgically treated ASH (Baucher G et al., Predictive Factors of Poor Prognosis After Surgical Management of Traumatic Acute Subdural Hematomas: A Single-Center Series. World Neurosurg. 2019). The high rate of ncSE in this patient series (13.7%, 18/131) still surprises me, as does the 100% mortality rate of this subgroup of patients. This being said, at least the authors have the merit to share honest results regarding functional outcome.

Minor comments

2. Materials and methods

2.1 Inclusion criteria

“We defined minor-trauma 53 head injury as a direct head impact without prolonged loss of consciousness

The following recommendations could have been used to define this inclusion criteria more precisely:

- Masters SJ, McClean PM, Arcarese JS, Brown RF, Campbell JA, Freed HA, Hess GH, Hoff JT, Kobrine A, Koziol DF, et al. Skull x-ray examinations after head trauma. Recommendations by a multidisciplinary panel and validation study. N Engl J Med. 1987 Jan 8;316(2):84-91. doi: 10.1056/NEJM198701083160205. PMID: 3785359.

- Marshall S, Bayley M, McCullagh S, Velikonja D, Berrigan L, Ouchterlony D, Weegar K; mTBI Expert Consensus Group. Updated clinical practice guidelines for concussion/mild traumatic brain injury and persistent symptoms. Brain Inj. 2015;29(6):688-700. doi: 10.3109/02699052.2015.1004755. Epub 2015 Apr 14. PMID: 25871303.

3. Results

3.1 Semantic homogeneity

The authors should decide whether they use first author’s name or hospital’s name in the article.

The hospital’s name is used in Fig. 2, Table 2 and Table 3 whereas the first author’s name is used in the rest of the manuscript. This is confusing.

3.2 Table 2 and 3

I don’t understand GOS 1-5 and GOS 6. GOS ranges between 1 and 5, 4 and 5 being “good functional outcome”. If the authors want to compare the 30-days mortality depending on the SHE score, the groups should be “GOS 1” and “GOS 2-5”

3.2 Table 7

SHE score 2 is missing.

4. Conclusion

Once again, I think that the authors should temper their findings regarding post-operative ncSE. EEG is part of the standard paraclinical protocol of every neurosurgical center in case of unresponsive patient in the early postoperative period.

In my opinion, ncSE should rather be highlighted as a supplementary risk factor of dismal prognosis after surgical evacuation of ASH. Because of the lack of power for this subgroup analysis (18 patients) which should be noted one way or another, this could open the way to future more powerful studies aiming to analyze the impact of ncSE on the 30-days mortality after surgical evacuation of ASH.

Conclusion of the review

Although this is a retrospective study, the authors have performed a well-designed external validation work for the SHE score. This being said, a few points should be revised before this work can be accepted for publication.

I would be pleased to review this manuscript once again in the near future !

Review decision

Major revision

Reviewer 2 Report

The authors present an external validation of a previously introduced score to predict 30-day mortality in patients >65 years undergoing surgery for SDH, and add to this also an investigation on the role of status epilepticus on 30-say mortality.

The results confirm the findings from the original paper, and the authors further observe that status epilepticus is a major predictor of 30-day mortality, and that surgery is associated with epilepsy in patients with aSDH.

Major concerns pertain the novelty of the findings, as the authors susbtantially confirm previous findings with a smaller cohort than the original paper. Also, stating that epilepsy is associated with higher mortality and that surgery is associated with epilepsy may not need a specific paper to sustain this.

Minor observation concern the definition of the SHE-score: in the introduction it seems like the authors made some confusion between aSDH and cSDH, and they report a wrong definition of SHE score: in the original paper, both aSDH and cSDH were considered. A later paper dealt precisely with cSDH prediction with SHE (Luca Petrella, Giovanni Muscas, Vita Maria Montemurro, Giancarlo Lastrucci, Enrico Fainardi, Gastone Pansini, Alessandro Della Puppa,Use of the Subdural Hematoma in the Elderly (SHE) Score to Predict 30-Day Mortality After Chronic Subdural Hematoma Evacuation,World Neurosurgery,Volume 157,2022,Pages e294-e300.)

Round 2

Reviewer 1 Report

Thank you for reaching out for the second round of reviewing.

The authors properly addressed the majority of my comments. This being said, I still have a few concerns which require their attention.

General remarks

First, I want to say that I greatly appreciate the authors’ reactivity in conducting the corrections of this manuscript. Most of my comments have been addressed.

I think that this work will be of some help in the surgical decision making in face with an acute subdural hematoma in the elderly. This being said, my current feeling is the lack of coherence / order in the manuscript, which renders its full reading quite difficult! One of the main points of this review is to encourage the authors to reorganize their manuscript.

The other striking observation is the omnipresence of interpretation of the consequences of nonconvulsive status epilepticus (ncSE) in the whole manuscript. ncSE constitutes a secondary endpoint. I understand the will of the authors to highlight this finding, but it should remain a secondary endpoint!

Major comments

1. Title

I understand the will of the authors to keep their findings regarding nonconvulsive status epilepticus (ncSE) in the title.

Nevertheless, I will stick to my previous comment. The title is not the proper place to highlight a secondary endpoint, regardless of the importance of this endpoint. The authors will not lose the credit of this finding whether it appears in the title or not.

2. Methods

2.1 Primary and secondary endpoints

Even if the order of the paragraphs in the Methods’ section has been modified, the wording of these paragraphs is still confusing.

Perhaps the number of paragraphs in the Methods’ section could be reduced. Besides, the expressions “primary endpoint” and “secondary endpoint” should appear clearly in order to ease the reading of the Methods’ section.

3. Results

3.1 Tables

The following tables should be combined in the same table:

-Table 1 (patient characteristics)

-Table 6 (Comparison between patients who underwent surgical evacuation vs. those with conservative treatment).

The following tables should be combined in the same table:

-Table 3 (Comparison Vychopen vs. Alfort, 30-day mortality. SHE: subdural hematoma in elderly score),

-Table 5 (Distribution of non-convulsive Status epilepticus in correlation with SHE score Nonconvulsive status epilepticus (n) of total (%), SHE: subdural hematoma in elderly score, n: number of patients.)

3.2 Comparability with Alford’s study population

If comparison between the study population in this manuscript and the study population in Alford’s original work is not possible, it should appear in the limitations paragraph.

3.3 ncSE

The authors’ response to my previous comment 3.4:

“The analysis of the score was retrospective. In this dataset, we did not use SHE score for surgical indication. Nevertheless, if divided in low SHE score (SHE 0 – 2) and high SHE score (SHE 3 – 4), there is no statistical difference between the both groups, (p = 0.3998).”

“In our data, the insertion of subdural drain was not significantly associated with development of ncSE. Therefore this was not addressed in the multivariate analysis (p = 0.2039”

Should appear in the Discussion section in order to give to the reader a better understanding of the surgical indications and factors associated or not with ncSE.

3.4 3.5 Institutional habits regarding epilepsy management

Given the importance of the authors’ findings regarding the consequences of ncSE (mortality rate 100%), I think that the authors’ institutional protocol regarding the prophylactic use of anti-epileptic drug (German neurosurgical society) should be clearly explained and critically confronted to recent evidence:

- Won SY, Konczalla J, Dubinski D, Cattani A, Cuca C, Seifert V, Rosenow F, Strzelczyk A, Freiman TM. A systematic review of epileptic seizures in adults with subdural haematomas. Seizure. 2017 Feb;45:28-35. doi: 10.1016/j.seizure.2016.11.017. Epub 2016 Nov 25. PMID: 27914224.

This may constitute a supplementary limitation in the analysis of the postoperative occurrence of ncSE.

4. Discussion

4.1 SHE score

The main finding of this manuscript is the external reproducibility of the SHE score.

This could be compared with the intracerebral hemorrhage score (ICH score) which is of similar use in an emergency setting in face with intracerebral hemorrhage, and is widely used in a day-to-day clinical practice:

-Hemphill JC 3rd, Bonovich DC, Besmertis L, Manley GT, Johnston SC. The ICH score: a simple, reliable grading scale for intracerebral hemorrhage. Stroke. 2001 Apr;32(4):891-7. doi: 10.1161/01.str.32.4.891. PMID: 11283388.

4.2 ncSE

I have read the authors’ response to my comments 4.1 and 4.2.

Nevertheless, a striking fact when I read the discussion section once again is the omnipresence of statements concerning ncSE, whereas it constitutes the secondary endpoint here:

-“Additionally, we identified development of non-convulsive status epilepticus postoperatively as an independent risk factor associated with 100 % mortality regardless of the initial SHE score.”

-“Regarding surgery as a risk factor for SE, the appropriate indication to hematoma evacuation seems to play a crucial role in aSDH therapy”

-“However, despite the good outcome the development of non-convulsive SE is a prognosis-limiting factor that should lead to careful preoperative evaluation and accurate operative indication”

-“As previously described, the development of ncSE dramatically increases the risk of a poor cognitive and neurological outcome in adults [24], as well as in neurological critical care patients with ICH”

The authors should discuss the potential causes and consequences of ncSE in the second paragraph of the Discussion section once and for all, because it is the secondary endpoint. What is more, they should temper their findings because they may be biased with the subgroup analysis including only 18 patients, and the absence of systematic prophylactic anti-epileptic medication in patients with acute subdural hematoma.

5. Conclusion

5.1 Interpretation of the SHE score

In the conclusion, the authors state that:

“The possible implication of our clinical finding might be a careful consideration of therapy goal by patients with high SHE score (3-4) who postoperatively develop ncSE.”

I think that they should divide this conclusion in two parts. First, the interpretation of the SHE score which may help the neurosurgeon in the emergency setting to not perform surgery for patients with SHE score of 3 or 4 and major medical comorbidities.

Second, the interpretation of the consequences of ncSE, which may assist the multidisciplinary team’s decision to limit resuscitation care in patients suffering from ncSE after surgical evacuation of acute subdural hematoma.

From my point of view, this distinction is important because these elements constitute separate endpoints, and they do not occur at the same time during the patient care.

I appreciate the authors’ addition:

“Our data suggest 100% mortality in this subgroup of patients. The major limitation of this statement is the lack of statistical power (in total, only 18 patients with ncSE were identified).

Minor comments

1. Materials and methods

1.1 Surgical protocol

The authors state that:

“After hematoma evacuation, up to 2 subdural drains might be placed to drain the remaining blood”

One could add “[…] based on the surgeon’s judgement. The amount of fluid collected in these drains was not systematically noted” 

Conclusion of the review

I appreciate the efforts that the authors have put in the revision of this manuscript.

There is still room for improvement, though. I hope the authors will find my comments useful in order to do so.

I will be pleased to review this work once again in the very near future!

Reviewer 2 Report

As stated in thre previous round fo review, is this Reviewer's opinion that the paper needs substanial and basic revisions.

As far as it can be evaluated, this revisions does not add any substantial change to the previous version, at least concernign this Reviewer's comments.
